# Isolation of an Obligate Mixotrophic Methanogen That Represents the Major Population in Thermophilic Fixed-Bed Anaerobic Digesters

**DOI:** 10.3390/microorganisms8020217

**Published:** 2020-02-06

**Authors:** Misa Nagoya, Atsushi Kouzuma, Yoshiyuki Ueno, Kazuya Watanabe

**Affiliations:** 1School of Life Sciences, Tokyo University of Pharmacy and Life Sciences, Hachioji, Tokyo 192-0392, Japan; s158048@toyaku.ac.jp (M.N.); akouzuma@toyaku.ac.jp (A.K.); 2Kajima Technical Research Institute, Chofu, Tokyo 182-0036, Japan; uenoyo@kajima.com

**Keywords:** methanogenesis, acetoclastic methanogenesis, hydrogenotrophic methanogenesis, mixotrophy, carbon-monoxide dehydrogenase, acetyl-CoA synthase, *Methanothermobacter*

## Abstract

*Methanothermobacter* Met2 is a metagenome-assembled genome (MAG) that encodes a putative mixotrophic methanogen constituting the major populations in thermophilic fixed-bed anaerobic digesters. In order to characterize its physiology, the present work isolated an archaeon (strain Met2-1) that represents Met2-type methanogens by using a combination of enrichments under a nitrogen atmosphere, colony formation on solid media and limiting dilution under high partial pressures of hydrogen. Strain Met2-1 utilizes hydrogen and carbon dioxide for methanogenesis, while the growth is observed only when culture media are additionally supplemented with acetate. It does not grow on acetate in the absence of hydrogen. The results demonstrate that *Methanothermobacter* sp. strain Met2-1 is a novel methanogen that exhibits obligate mixotrophy.

## 1. Introduction

Methanogens are archaea that produce methane as the catabolic end product [1]. In nature, methanogens play important roles in organic-matter decomposition under anaerobic conditions [1]. In addition, they are essential members of microbiomes in anaerobic digesters that convert organic wastes into fuel gas [2]. It is known that two types of methanogens mainly contribute to anaerobic digestion, one that produces methane from carbon dioxide using hydrogen/formate as reducing powers (hydrogenotrophic methanogen), and the other that produces methane from acetate (acetoclastic methanogen) [1].

Thermophilic fixed-bed anaerobic digesters (TFDs) facilitate high-rate conversion of organic wastes into methane [3], and commercial TFDs are operated in Japan and other countries for the treatment of food and brewery wastes [4]. Recently, in order to characterize microbes that contribute to the efficient methanogenesis in TFDs, microbiomes established in laboratory-scale TFDs were analyzed by metagenomic and metatranscriptomic approaches [5]. It has been found that metagenome-assembled genomes (MAGs) Mes1 and Met2 represent two major populations that account for over 30% and 20%, respectively, of the total biofilm populations in TFD [5]. Among them, Mes1 exhibits substantial similarities to *Methanosarcina thermophila* [6] in terms of phylogeny, gene contents and genome structures. On the other hand, although Met2 is considered to be affiliated with the genus *Methanothermobacter* based on 16S rRNA- and *mcrA*-gene sequences, substantial differences are found in gene contents and genome structures between Met2 and authentic *Methanothermobacter* strains, such as *Methanothermobacter thermautotrophicus* ΔH [7]. In particular, reconstructed catabolic pathways indicate that the Met2 MAG does not code for a carbon-monoxide dehydrogenase/acetyl-CoA synthase complex (CODH/ACS), one of the essential enzymes in the methanogenic pathways [8]. This finding suggests the possibility that Met2 is incapable of utilizing the acetyl-CoA pathway for CO_2_ fixation and also of producing methane from acetate. It is therefore considered that Met2 utilizes CO_2_/H_2_ only for conserving energy by methanogenesis, while acetate is used as a carbon source. Based on these assumptions, Met2 is likely a novel obligate mixotrophic methanogen that is entirely different from autotrophic hydrogenotrophic methanogens belonging to the genus *Methanothermobacter*. Furthermore, obligate mixotrophy is not well appreciated for methanogens. It is also noteworthy that, in the TFD biofilm, Met2 outgrows Met20, another reconstructed MAG that constitutes 0.2% of the total biofilm populations. Met20 is closely related to *M. thermautotrophicus* and codes for CODH/ACS [5]. We deduce that Met2-type methanogens have ecological advantages over other hydrogenotrophic methanogens in TFD biofilms, and phenotypes associated with the lack of CODH/ACS would be relevant.

Given the physiological perspectives on MAG Met2 inferred from the metagenomic and metatranscriptomic analyses [5], an archaeon that represents Met2-type methanogens was isolated and examined for its physiological properties. The primary purpose of the present work was therefore to isolate archaeal strains corresponding to Met2. Phylogenetic and physiological characteristics of an isolate is reported in this article.

## 2. Materials and Methods

### 2.1. Isolation Source and Sampling

A laboratory-scale TFD was operated at 55 °C as described previously [5]. When methane was stably produced at an organic-loading rate of approximately 3 grams liter^−1^ day^−1^, biofilm was sampled by inserting a syringe into the digester and immediately spiked into butyl rubber-sealed anaerobic vials containing medium 119 (Deutsche Sammlung von Mikroorganismen und Zellkulturen GmbH, Braunschweig, Germany) and either H_2_/CO_2_ (80%/20%) or pure nitrogen gas.

### 2.2. Cultivation in Liquid Media

Liquid cultivation was conducted in butyl rubber-sealed vials (110 mL in capacity) containing 50 mL of medium 119 at 65 °C. Modified 119 media were also used, and compositions of these media were specified where necessary. The head space in a vial was filled either with pure nitrogen gas or with H_2_/CO_2_ gas. A vial was inoculated with a microbial suspension using a syringe, after a vial was sealed with a butyl-rubber cap and aluminum crimp. Microbial cells in liquid media were counted using a fluorescence microscope (BX60, Olympus, Tokyo, Japan), after cells were sampled using a syringe and stained with 4′,6-diamidino-2-phenylindole (DAPI) [9]. For physiological characterization of an isolate, a culture vial prepared as described above was inoculated with 0.5 mL of a pre-grown culture, and the growth was analyzed as described above. Purity of a culture was checked by repeated microscopic and sequencing analyses.

### 2.3. Cultivation on Solid Media

Solid slant media (30 mL in volume) were formed in butyl rubber-sealed vials (50 mL in capacity) using gellan gum (0.9%) (Wako Pure Chemicals, Osaka, Japan) as a solidification agent. Slants contained medium 119 supplemented with ampicillin (100 mg L^−1^) and vancomycin (200 mg L^−1^), and the head-space gas was H_2_/CO_2_. Cell suspension was streaked on the surface of solid media using a syringe, after vials were sealed with butyl-rubber caps and the head-space gas was replaced with H_2_/CO_2_. Cells were grown at 65 °C, picked using needles in an anaerobic chamber (Bactron, Sheldon Manufacturing, Cornelius, OR, USA) and immediately inoculated into liquid media in vials.

### 2.4. Gas Analyses

Methane, hydrogen, nitrogen, and carbon dioxide in the head space of a vial were measured using a gas chromatograph (GC-14A, Shimadzu, Kyoto, Japan) equipped with a thermal conductivity detector and a molecular sieve 5A 60–80/Porapack Q 80–100 column (Shimadzu). The column, injection, and detector temperatures were 50 °C, 100 °C, and 80 °C, respectively [10].

### 2.5. Phylogenetic Analyses

DNA was extracted from a liquid culture using a Fast DNA Spin Kit for Soil (MP Bio Japan, Tokyo, Japan). Amplification of a 16S rRNA gene fragment was conducted using primers 515F and 1492R according to methods described elsewhere [11]. Amplification of a *mcrA* gene fragment was conducted using primers mcr1F and mcr1R according to methods described previously [12]. PCR products were purified using a QIAquick PCR purification kit (Qiagen, Tokyo, Japan) and sequenced by a standard procedure. Nucleotide sequences were analyzed using the Blast program (NCBI, http://www.ncbi.nlm.nih.gov/), and phylogenetic trees were constructed using a MEGA5 software [13].

Nucleotide sequences determined in the present study are deposited in the DDBJ, EMBL, and NCBI nucleotide sequence databases under accession numbers LC514466 to LC514469.

## 3. Results and Discussion

### 3.1. Colony Isolation

Vials containing medium 119 and H_2_/CO_2_ were inoculated with the TFD biofilm suspension and incubated at 65 °C for promoting the growth of methanogens. The present study employed 65 °C, since thermophilic *Methanosarcina*, such as *Ms. thermophila* are unable to grow at this temperature [14]. The medium contained yeast extract, acetate, formate, sludge fluid and a fatty-acid mixture as organic ingredients. After methane was detected in the head space, the culture was spread onto the solid slant media containing medium 119. Colonies were formed several days after commencing the cultivation concomitant with the detection of methane in the head space. Several colonies were transferred to the liquid media, and resultant methanogenic cultures were used for the sequencing of 16S rRNA and *mcrA* genes. It was found that *mcrA* sequences determined for colonies were 100% identical to that of MAG Met20 [5], but not of Met2. Besides, 16S rRNA gene sequences for colonies were 100% identical to that of *M. thermautotrophicus* ΔH. We were unable to compare 16S rRNA gene sequences of the colonies to that of Met20 deposited in the database, since there was no overlap between them. It should however be noted that the 16S rRNA gene sequence of Met20 is 100% identical to that of strain ΔH [5].

One of the isolates is named strain Met20-1. Microscopic observation showed that cells of strain Met20-1 were extended rods (Figure 1A), similar to those reported previously for strain ΔH [10]. Since our previous work detected MAG Met20 as a minor population (0.2% of the total biofilm populations) in TFD biofilms [5], this minor methanogen outcompeted Met2 under the high hydrogen partial pressure in the initial enrichment culture and eventually isolated.

In the next trial, methanogens were enriched from the TFD biofilm in medium 119 under the nitrogen atmosphere to suppress the growth of methanogens that rapidly grow under the H_2_/CO_2_ atmosphere (e.g., strain Met20-1). It was expected that acidogenic bacteria, such as those represented by MAG *Coprothermobacter* Cop3 [5], would produce hydrogen and acetate and promote the syntrophic growth of Met2 methanogens as was the situation deduced for microbiomes in TFD biofilms. Methanogenic enrichment cultures were streaked onto the slant media, and colonies formed on these slants under the H_2_/CO_2_ atmosphere were grown in the liquid media for determining sequences of 16S rRNA and *mcr* genes. The 16S rRNA gene analyses revealed the presence of bacteria closely related to *Coprothermobacter proteolyticus* [15], while the *mcrA* gene analyses detected the Met2 sequence. In addition, two different morphotypes of cells were observed under the microscope in liquid cultures (Figure 1B), and it was considered that cocci were *Coprothermobacter* cells, while filaments were *Methanothermobacter* cells related to Met2. Studies have shown that *Coprothermobacter* and *Methanothermobacter* grow under hydrogen-mediated syntrophic interactions and form tight co-aggregates [16]. It was therefore considered that the separation of these two microbes by colony formation on solid media would be difficult.

### 3.2. Limiting Dilution

It has been reported that the growth of *Coprothermobacter* is suppressed in the presence of hydrogen gas at high partial pressures [16]. In order to eliminate *Coprothermobacter* cells from the Met2/*Coprothermobacter* mixed culture (Figure 1B), we employed a limiting-dilution approach under the H_2_/CO_2_ (80%/20%) atmosphere. In this approach, we eliminated the fatty-acid mixture and sludge fluid from medium 119, since these were considered to be potential substrates for *Coprothermobacter*. A methanogenic culture at the highest dilution was spread onto the slant medium, and colonies formed on the slants were cultivated in the modified 119 liquid medium for the phylogenetic and microscopic analyses. It was found that 16S rRNA and *mcrA* sequences determined for the colonies were 100% identical to those of MAG Met2. In addition, the microscopic observation showed that cells were solely filamentous, and coccal cells were not present (Figure 1C). From these results, we conclude that a methanogenic archaeon that represents Met2-type methanogens is successfully isolated, and the isolate is named Met2-1. It is interesting that cells of strain Met2-1 are much longer than those of strain Met20-1 (Figure 1). The isolation of strain Met2-1 allows us to investigate physiological features of methanogens that constitute major populations in TFDs.

### 3.3. Phylogenetic Characteristics

Sequences for 16S rRNA and *mcrA* genes were used to phylogenetically characterize strain Met2-1 (Figure 2). As shown in these figures, this archaeon is affiliated with the genus *Methanothermobacter*, while it is relatively distantly related to authentic *Methanothermobacter* strains, such as *M. thermautotrophicus* ΔH. On the other hand, it is shown that strain Met2-1 is closely related to two methanogens, *Methanothermobacter tenebrarum* and *Methanothermobacter crinale.* Percent identities in the partial 16S rRNA gene sequence of strain Met2-1 to those of *M. tenebrarum*, *M. crinale*, *M. thermautotrophicus* are 99.7%, 99.6%, and 96.6%, respectively, while those in the partial *mcrA* gene sequence are 92.9%, 93.2%, and 82.7%, respectively.

*M. tenebrarum* was isolated from formation water in a natural gas field [17], while *M. crinale* was isolated from oil sands sampled from a high-temperature underground oil reservoir [18]. We are interested in the fact that the dominant methanogen in the high-temperature anaerobic digester (strain Met2-1) is more closely related to thermophilic methanogens that thrive in subsurface oil/gas fields than to those found in anaerobic digesters.

### 3.4. Growth Characteristics

From the metagenomic data reported in our previous study [5], it was predicted that strain Met2-1 is an obligate mixotrophic methanogen that does not possess CODH/ACS and requires both acetate and H_2_/CO_2_ for growth. In order to address this hypothesis, strain Met2-1 was grown in three different culture systems, namely, acetate plus H_2_/CO_2_, only H_2_/CO_2_ and only acetate, and growth trends of Met2-1 were compared with those of the autotrophic hydrogenotrophic methanogen, *M. thermautotrophicus* ΔH (Figure 3). In this analysis, we used a modified 119 medium that contained neither formate, fatty-acid mixture nor sludge fluid. In addition, the amount of yeast extract was decreased from 1.0 to 0.1 g L^−1^.

Figure 3 shows that, while a substantial amount of methane is produced by strain Met2-1 in the absence of acetate, its growth is largely suppressed without acetate. Strain Met2-1 slightly grows on H_2_/CO_2_ in the absence of acetate, while this growth is considered to be dependent on some organics in yeast extract. It is shown that the growth trends of strain Met2-1 are entirely different from those of ΔH, since methanogenesis and growth of strain ΔH are not affected by acetate. Both strains did not grow on acetate (in the absence of H_2_/CO_2_). These results support the idea that strain Met2-1 is an obligate mixotrophic methanogen that requires both H_2_/CO_2_ and acetate for growth. 

It has been reported that the two *Methanothermobacter* strains closely related to strain Met2-1, *M. tenebrarum* and *M. crinale*, are hydrogenotrophic methanogens, while acetate is required for their growth [17,18]. Among them, a draft genome is available for *M. tenebrarum*, while *M. crinale* has not yet been sequenced. Although a solid conclusion cannot be obtained with a draft genome, genes coding for CODH/ACS are not found in *M. tenebrarum*. On the other hand, genomes of other *Methanothermobacter* strains, including *M. marburgensis*, *M. thermautotrophicus* and *M. wolfeii* code for CODH/ACS. 

## 4. Conclusions

The present work isolated *Methanothermobacter* sp. Met2-1 that represents the dominant populations in TFDs. It is demonstrated that strain Met2-1 requires both H_2_/CO_2_ and acetate for growth and represents obligate mixotrophic methanogens. This strain is considered to be affiliated with the genus *Methanothermobacter*, while it is closely related to some methanogens isolated from underground oil fields.

From the results reported herein, it is suggested that methanogens represented by Met2-1 grow under syntrophic interactions with acidogenic bacteria, such as *Coprotermobacter*, where acetate, H_2_ and CO_2_ transfer between these microbes. It is likely that these methanogens constitute the major population in TFDs with the ability to efficiently establish the syntrophic interactions.

A prominent feature of the present work is that the isolation procedure developed in the work is useful for isolating methanogens that thrive under syntrophic interactions with acidogenic bacteria. In the procedure, special attention is paid to suppress methanogens that rapidly grow under high partial pressures of hydrogen and to eliminate acidogenic bacteria from syntrophic cultures. Since the procedure would be widely applicable to isolating ecologically important methanogens from a variety of anaerobic environments, we expect that our work will advance our understanding of the ecology and physiology of yet uncultured methanogens in the natural and engineered environments.

## Figures and Tables

**Figure 1 microorganisms-08-00217-f001:**
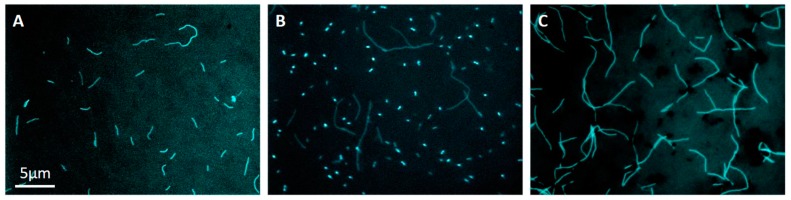
Fluorescent micrographs of DAPI-stained cells grown from colonies on sold media under methanogenic conditions. (**A**) Cells cultivated from a colony formed on slants containing medium 119 after methanogens were enriched under the H_2_/CO_2_ gas (strain Met20-1). (**B**) Cells from a colony formed on slants containing medium 119 after methanogens were enriched under the nitrogen gas. (**C**) Cells from a colony formed on slants containing the modified 119 medium after limiting dilution under the H_2_/CO_2_ gas (strain Met2-1). (**A**) Bar (5 μm) in panel (**A**) also applies to panels (**B**,**C**).

**Figure 2 microorganisms-08-00217-f002:**
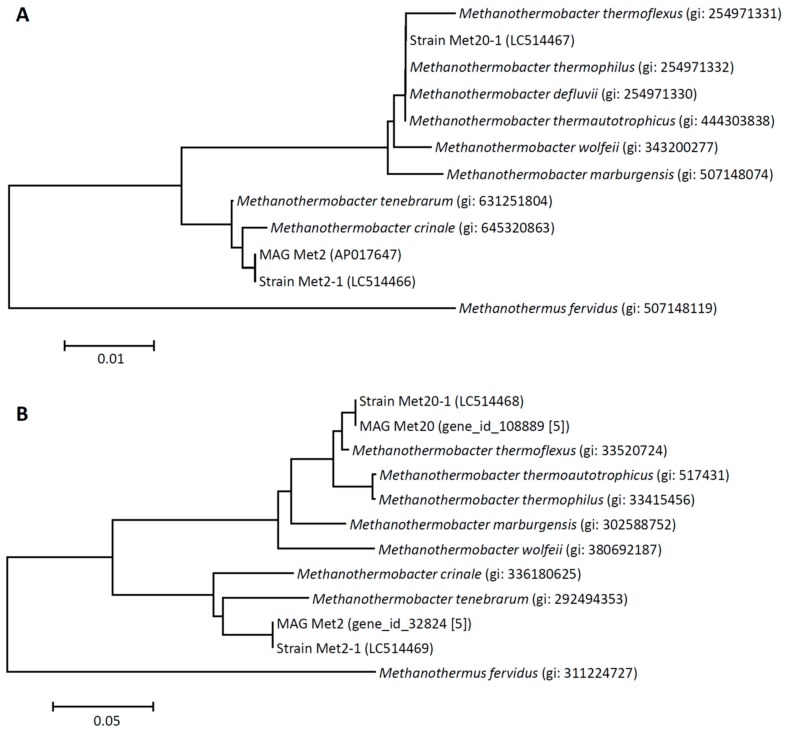
Phylogenetic trees based on 16S rRNA gene sequences (**A**) and *mcrA* sequences (**B**) showing phylogenetic relationships among archaea affiliated with the genus *Methanothermobacter*. *Methanothermus fervidus* DSM 2088 is used as the outgroup. Bootstrap values (100 trials, only values greater than 50 are shown) are indicated at branching points. Sequence divergences are indicated with bars, and accession numbers are shown in parentheses.

**Figure 3 microorganisms-08-00217-f003:**
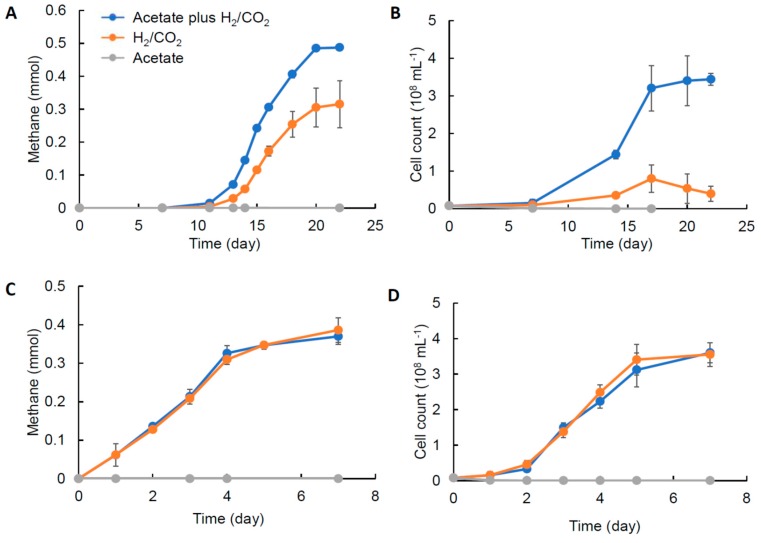
Methanogenesis (**A**,**C**) and growth (**B**,**D**) of *Methanothermobacter* sp. Met2-1 (**A**,**B**) and *M. thermautrotrophicus* ΔH (**C**,**D**) on three different substrates.

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
