# Peer review of "Isolation of an Obligate Mixotrophic Methanogen That Represents the Major Population in Thermophilic Fixed-Bed Anaerobic Digesters"

_microorganisms, 2020, doi:10.3390/microorganisms8020217_

Round 1

Reviewer 1 Report

Review MDPI microorganisms

Manuscript entitled: “Isolation of an obligate mixotrophic methanogen that represents the major population in thermophilic fixed-bed anaerobic digesters”

Authors: Misa Nagoya, Atsushi Kouzuma, Yoshiyuki Ueno and Kazuya Watanabe

Review

Dear Misa Nagoya, Dear Atsushi Kouzuma, Dear Yoshiyuki Ueno, Dear Kazuya Watanabe, Dear respected authors,

It is with great interest and pleasure that I have reviewed your manuscript of which the details are indicated above and that was submitted to MDPI microorganisms recently.

Efforts to isolate microorganisms of which the existence was indicated by shotgun sequencing approaches should in my opinion be encouraged especially when interesting capabilities have been found from the assembled MAGs. This manuscript is exemplary of such an effort and I congratulate you with your achievement.

In my opinion this effort deserves to be reported well and although I am very positive about your work, I do think your manuscript should be adapted to express better the finding to an audience of interested readers. Please consider carefully and take time to address my comments below to adapt your manuscript. I hope you can re-submit the adapted version of this manuscript so it can be considered for publication in MDPI microorganisms.

Thank you and best wishes!

Comments

C1: In the abstract (line 10) the genus name Methanothermobacter should be spelled accordingly, please check and remove the extra “o”.

C2: Please re-consider the keywords. Why add Coprothermobacter rather than Methanothermobacter or should both be added? And how about the mixotrophy? Please update the keywords or carefully consider the presence of each of them again, are they all relevant?

C3: In your manuscript you suggest the possibility that methanogens represented by Met2 are “incapable of using CO2 as carbon source and also of producing methane from acetate”. This was based on the absence of a carbon-monoxide dehydrogenase / acetyl-CoA synthase complex. Doesn’t the absence of this particular synthase complex just indicate that the strain is not an acetoclastic methanogen? I don’t understand how we can assume about the carbon source as in my opinion with the absence of the synthase complex it is also possible to use the carbon from CO2 as a carbon source.

C4: To make more clear a key point of this manuscript, the introduction should have at least a few lines more explanation on the finding of mixotrophy in methanogenesis. It deserves these extra lines as the concept is very interesting and has to be communicated more clearly.

C5: What are the methods for the physiological growth tests of the isolated strain Met2-1?

C6: Was the isolated strain Met2-1 also made publicly available by submission in a culture collection?

C7: The information in the section 3.1 and 3.2 is very interesting, but can it be slightly adjusted to be more focused on the relevant observations and the main finding of the Met2-1 isolation?

C8: To make more clear the phylogenetic relatedness the percent identity of the 16S rRNA sequence from the subject to some target organisms can be used. How many percent are for example the 16S rRNA and mcrA sequences of Met2-1 identical to deltaH, M. tenebrarum and M. crinale?

C9: There is a genome sequence at least for the M. tenebrarum. Please include the presence of the genome sequences and check to confirm the absence of CODH/ACS as speculated in line 174. Perhaps also interesting to do this check for the other Methanothermobacter genomes available (?).

C10: The conclusions are rather general. Please re-consider the conclusions. Perhaps the more specific phylogenetic analysis and physiological growth tests can be included to make this less general and perhaps some further interesting characteristics of the strain Met2-1 can be added.

Author Response

Thanks a lot for valuable comments on our manuscript. Please check our responses below (described after >).

Reviewer 1

Comments

C1: In the abstract (line 10) the genus name Methanothermobacter should be spelled accordingly, please check and remove the extra “o”.

> This is revised (L10 in the revised manuscript).

C2: Please re-consider the keywords. Why add Coprothermobacter rather than Methanothermobacter or should both be added? And how about the mixotrophy? Please update the keywords or carefully consider the presence of each of them again, are they all relevant?

> Some keywords are changed (L19 to L20).

C3: In your manuscript you suggest the possibility that methanogens represented by Met2 are “incapable of using CO2 as carbon source and also of producing methane from acetate”. This was based on the absence of a carbon-monoxide dehydrogenase / acetyl-CoA synthase complex. Doesn’t the absence of this particular synthase complex just indicate that the strain is not an acetoclastic methanogen? I don’t understand how we can assume about the carbon source as in my opinion with the absence of the synthase complex it is also possible to use the carbon from CO2 as a carbon source.

> Acetoclastic methanogenesis is solely dependent on the synthase complex; the absence of this synthase complex therefore indicates that the strain is not an acetoclastic methanogen. On the other hand, although the synthase complex constitutes the major CO2 fixation route, CO2 fixation is also possible with several other mechanisms, including pyruvate carboxylate (Samuel et al. 2007). However, pyruvate should be synthesized from acetate via acetyl-CoA in the absence of the synthase complex, suggesting that acetate is indispensable for methanogens that do not have the synthase complex. We therefore rephased the sentence (L44 to L45).

C4: To make more clear a key point of this manuscript, the introduction should have at least a few lines more explanation on the finding of mixotrophy in methanogenesis. It deserves these extra lines as the concept is very interesting and has to be communicated more clearly.

> A sentence is added (L49 to L50).

C5: What are the methods for the physiological growth tests of the isolated strain Met2-1?

> A sentence was added for describing methods (L74 to L76).

C6: Was the isolated strain Met2-1 also made publicly available by submission in a culture collection?

> Not yet. We will deposit it to culture collections after some more examination for taxonomic characterization will be conducted.

C7: The information in the section 3.1 and 3.2 is very interesting, but can it be slightly adjusted to be more focused on the relevant observations and the main finding of the Met2-1 isolation?

> A sentence is added to describe the importance of the Met2-1 isolation (L148 to L149).

C8: To make more clear the phylogenetic relatedness the percent identity of the 16S rRNA sequence from the subject to some target organisms can be used. How many percent are for example the 16S rRNA and mcrA sequences of Met2-1 identical to deltaH, M. tenebrarum and M. crinale?

> These data are described (L157 to L160).

C9: There is a genome sequence at least for the M. tenebrarum. Please include the presence of the genome sequences and check to confirm the absence of CODH/ACS as speculated in line 174. Perhaps also interesting to do this check for the other Methanothermobacter genomes available (?).

> Not found in M. tenebrarum. However, we are unable to conclude the deficiency of CODH/ACS in M. tenebrarum with the draft genome. CODH/ACS genes are found in other Methanothermobacter genomes. This information is added (L184 to L188).

C10: The conclusions are rather general. Please re-consider the conclusions. Perhaps the more specific phylogenetic analysis and physiological growth tests can be included to make this less general and perhaps some further interesting characteristics of the strain Met2-1 can be added.

> Descriptions are added to explain characteristics of strain Met2-1 (L190 to L194).

Reviewer 2 Report

This paper represents the isolation of Methanothermobactersp. Strain Met2-1, an important mixotrophic methanogen. The isolate itself and the data presented here add to the current knowledge of methanogens, and culture collections. 

I recommend the publication of this study some corrections:

Generally, more details are needed for Materials and Methods. Anaerobic cultivation is tricky. For instance, it is not clear how the authors transferred the colonies between solid and liquid media. How did they check the purity of the colonies?

Line 27. Remove “namely”

Line 62. Please correct to “organic-loading rate” not “organics-loading rate”.

Cultivation in liquid and solid media (Line 67-80). Have these been carried out under anaerobic conditions such as anaerobic chamber?

Line 102: Which solid media? The solid slant media as in Materials and Methods?

Line 153-155. What is the authors’ comment on this then?

Author Response

Thanks a lot for valuable comments on our manuscript. Please check our responses below (described after >).

Reviewer 2

Generally, more details are needed for Materials and Methods. Anaerobic cultivation is tricky. For instance, it is not clear how the authors transferred the colonies between solid and liquid media. How did they check the purity of the colonies?

> Some more explanations are added for colony transfer (L82 to L85) and purity check (L76 to L77).

Line 27. Remove “namely”

> Removed (L27).

Line 62. Please correct to “organic-loading rate” not “organics-loading rate”.

> Corrected (L63).

Cultivation in liquid and solid media (Line 67-80). Have these been carried out under anaerobic conditions such as anaerobic chamber?

> An anaerobic chamber was used only when cells are transferred between solid and liquid media, while the inside of culture vials was sufficiently anaerobic for the growth of methanogens. Some more description is added in the revised manuscript (L74 to L77).

Line 102: Which solid media? The solid slant media as in Materials and Methods?

> We specified this (L107 to L108).

Line 153-155. What is the authors’ comment on this then?

> The sentence is revised to more accurately indicate the meaning (L162 to L165).

Round 2

Reviewer 1 Report

Review MDPI microorganisms

Manuscript entitled: “Isolation of an obligate mixotrophic methanogen that represents the major population in thermophilic fixed-bed anaerobic digesters”

Authors: Misa Nagoya, Atsushi Kouzuma, Yoshiyuki Ueno and Kazuya Watanabe

Dear Misa Nagoya, Dear Atsushi Kouzuma, Dear Yoshiyuki Ueno, Dear Kazuya Watanabe, Dear respected authors,

It is with great interest and pleasure that I have reviewed your manuscript of which the details are indicated above and of which a revised version was submitted to MDPI microorganisms recently.

There are no further comments from my side and I will advise the editor to accept this revised paper for publication in MDPI microorganisms.

Thank you and best wishes!